# Effects of Multicomponent Exercise Training on the Health of Older Women with Osteoporosis: A Systematic Review and Meta-Analysis

**DOI:** 10.3390/ijerph192114195

**Published:** 2022-10-30

**Authors:** Diego Gama Linhares, Claudio Joaquim Borba-Pinheiro, Juliana Brandão Pinto de Castro, Andressa Oliveira Barros dos Santos, Luciano Lima dos Santos, Lilliany de Souza Cordeiro, Alexandre Janotta Drigo, Rodolfo de Alkmim Moreira Nunes, Rodrigo Gomes de Souza Vale

**Affiliations:** 1Postgraduate Program in Exercise and Sport Sciences, Rio de Janeiro State University, Rio de Janeiro 20550-900, Brazil; 2Laboratory of Exercise and Sport, Institute of Physical Education and Sports, Rio de Janeiro State University, Rio de Janeiro 20550-900, Brazil; 3Federal Institute of Pará, Pará State University, Pará 68459-876, Brazil; 4Stricto Sensu Post Graduate Program, São Paulo State University, São Paulo 13506-900, Brazil

**Keywords:** elderly, exercise, osteoporosis, bone density, quality of life, health, physical functional performance, postural balance, muscle strength, resistance training

## Abstract

This study aimed to analyze the effects of multicomponent exercise training in older women with osteoporosis. We conducted a systematic review following the PRISMA guidelines and registered on PROSPERO (number CRD42022331137). We searched MEDLINE (via PubMed), Web of Science, Scopus, and CINHAL databases for randomized experimental trials that analyzed the effects of physical exercise on health-related variables in older women with osteoporosis. The risk of bias in the studies was verified using the Cochrane Collaboration tool and the Jadad scale was used to assess the methodological quality of the studies. Fourteen randomized controlled trials were included, with a total of 544 participants in the experimental group and 495 in the control group. The mean age of all participants was 68.4 years. The studies combined two to four different exercise types, including strength, aerobic, balance, flexibility, and/or functional fitness training. The practice of multicomponent training with an average of 27.2 weeks, 2.6 sessions per week, and 45 min per session showed improvements in strength, flexibility, quality of life, bone mineral density, balance, and functional fitness and reduced the risk of falls in older women with osteoporosis. Multicomponent training was shown to be effective in improving health-related variables in older women with osteoporosis.

## 1. Introduction

The world population has shown an abrupt increase in older people in relation to the total population since the mid-twentieth century. Aging tends to be accompanied by a loss of bone and muscle mass and an increase in the percentage of fat due to the reduction of sex hormones, especially anabolic steroids [1,2].

In this sense, changes in bone mineral density (BMD) levels can generate classifications of osteoporosis, such as mild, moderate, and severe. About 200 million people have osteoporosis. In the next three decades, the number of people with this disease is expected to increase by up to three times. Women have a lower BMD and a higher risk of fractures from falls due to the reduction in estrogen and the occurrence of menopause. Other functional losses can occur, such as reduced strength and muscle mass, balance, and visual capacity, which can increase the risk of falls [3].

The risk of fractures increases with osteoporosis followed by morbidity due to the reduction of the bone mineral component. Mortality is directly related to increased hip fracture rates of around 20%. Other factors are related to osteoporosis, such as hypothyroidism and reduced calcitonin secretion, which can lead to reduced BMD. Adult women diagnosed with BMD below the mean of 2.5 (T-score), estimated by dual-energy X-ray absorptiometry (DXA), are more likely to fracture [4].

Fractures resulting from falls due to osteoporosis leave important sequelae, such as regular pain, musculoskeletal, respiratory, and postural dysfunctions, leading to low functional autonomy and quality of life (QoL). Effective treatments that include exercise are needed to reduce these physical risk factors, pain, and physical dysfunction [5].

Physical exercise can maintain bone mass, increase muscle strength, and improve balance. Thus, aerobic and/or resistance exercises alone, even when not associated with drug therapies, are effective in reducing the loss of BMD in women with osteoporosis [6,7]. Additionally, a training modality that involves different physical capacities in the same exercise session is defined as multicomponent training. The combination of different physical capacities in the same exercise session is used in this training to address older people’s functional needs and health. This favors adherence through the proposal of socialization and participation in group activities [8].

Hence, multicomponent training can improve perceptual-cognitive functioning and reduce the frailty state of older people, as the combination of strength, aerobic, balance, and/or flexibility exercises can increase muscle strength, power, balance, and flexibility. These factors are important, as they improve performance in activities of daily living (ADL) [9,10,11,12].

Senescence, lifestyle habits, and chronic pathologies can lead to sarcopenia, osteopenia, and osteoporosis. On the other hand, healthy aging with an active lifestyle can provide the older person with a better perception of QoL, ADL, and health [13,14]. Despite those benefits, the effects of multicomponent exercise training on the health of older women with osteoporosis are not clear. As osteoporosis is common in this population and multicomponent training shows high adhesion and adherence rates [8], it is important to know the effects of this modality in older women with osteoporosis.

In this sense, the present study is justified by the need to investigate the control of variables related to the prescription of exercises in older women with osteoporosis and the possible physiological effects that can be optimized with multicomponent training in obtaining important results on bone health. Therefore, the present study aimed to analyze the effects of multicomponent training on the health of older women with osteoporosis. We hypothesized that this training modality would be effective in improving health-related variables of older women with osteoporosis.

## 2. Materials and Methods

This study is characterized as a systematic literature review. The procedure for conducting this research followed the criteria of the preferred reporting items for systematic reviews and meta-analyses (PRISMA) [15]. This study was registered in the international prospective register of systematic reviews (PROSPERO), as number CRD42022331137.

### 2.1. Search Strategy

Two independent and experienced researchers conducted an electronic search without language or time filters, in April 2022, in the MEDLINE (via PubMed), Web of Science, Scopus, and CINHAL databases. Any disagreements between the two researchers were solved by consulting a third researcher. The terms related to the topic were osteoporosis, elderly, treatment, and physical exercise. Those terms and their synonyms were appropriately combined using the Boolean operators [OR], between synonyms, and [AND], between descriptors. Reproducible search strategies can be found in Appendix A.

### 2.2. Eligibility Criteria

Randomized controlled trials (RCTs) that analyzed the effects of multicomponent training that combines a minimum of two different exercise types (strength, aerobic, balance, flexibility, and/or functional fitness) on health-related variables in older women with osteoporosis were included. Studies that did not use physical exercise as the main intervention or osteoporosis as the main pathology, articles published in congress, systematic reviews, and meta-analyses were excluded.

### 2.3. Research Question

We based the research question and strategy of our study on the population, intervention, comparison, and outcome (PICO) model, often used in evidence-based practice and recommended for systematic reviews [16]. Therefore, the population was older women with osteoporosis, the intervention was multicomponent exercise training, the control was the group of participants that did not practice multicomponent exercise training, and the outcome was health-related variables. Therefore, the final PICO question was “What are the effects of multicomponent exercise training on health-related variables in older women with osteoporosis?”.

### 2.4. Risk of Bias Analysis

The risk of bias of each included RCT was assessed by the Cochrane Collaboration tool, available at: https://training.cochrane.org/handbook/, accessed on 10 April 2022. This tool consists of 7 domains: (1) generation of the random sequence; (2) allocation concealment; (3) blinding of evaluators and participants; (4) blinding of outcome evaluators; (5) incomplete outcomes; (6) reports of selective outcomes; (7) report on other sources of bias. Each domain has the risk of bias classified as “high”, “uncertain”, or “low”. The final score is assigned with the highest rating among the domains evaluated in each study [17]. Two authors independently performed the risk of bias assessment of each included study and a third researcher was consulted in case of divergences.

### 2.5. Methodological Quality Analysis

The Jadad scale was used for the analysis of the methodological quality of the RCTs. This instrument has 3 items with a total of 5 points: (1a) the study was described as randomized; (1b) the randomization was accurately performed; (2a) the study was a double-blind trial; (2b) the blinding was properly performed; (3) the study described the sample loss. The score can vary from 0 to 5. Studies with a score ≤ 3 are considered at high risk of bias. Two researchers conducted the methodological quality analysis. Any divergences in the analysis were sent to a third researcher for consensus [18].

### 2.6. Data Collection Process

Data from the included publications were independently extracted by two researchers. Disagreements were resolved in a consensus meeting with a third researcher. The following variables were extracted: authors, year of publication, country, characteristics of the study population (age, sample size, and BMD), and intervention data, including general and specific exercises, intervention duration (weeks), intensity and volume of training (duration of the training session, in minutes, and frequency, in times per week), evaluation, and outcomes for variables related to physical and mental health.

### 2.7. Meta-Analysis

We used the Review Manager 5.4.1 program, available at http://tech.cochrane.org/revman, accessed on 25 October 2022, to analyze the effects of multicomponent exercise training on the health of older women with osteoporosis. Meta-analyses were performed when two or more studies could be pooled. As variables were continuous, we used the inverse variance statistical method and the analysis model with the random effect. The effect measure was the difference between the means with a 95% confidence interval from the studies. The meta-analysis and distribution of the studies were analyzed by the weight of each variable in the meta-analysis.

### 2.8. Evidence Level Assessment

Two independent researchers used the grading of recommendations assessment, development and evaluation (GRADE) approach to evaluate the evidence level for each investigated outcome. The quality of evidence can be assessed by four classification levels: high, moderate, low, and very low. RCTs start with high quality of evidence, and observational studies begin with low quality of evidence. Five aspects can decrease the quality of the evidence: methodological limitations, inconsistency, indirect evidence, inaccuracy, and publication bias. Contrariwise, three aspects can increase the quality of the evidence: effect size, dose-response gradient, and confounding factor [19].

## 3. Results

In total, 919 studies were found following the proposed research methodology (MEDLINE via PubMed = 416; Scopus = 226; Web of Science = 108, CINAHL = 169). After using the selection criteria, 14 articles were included in the qualitative analysis and four studies provided data to be included in the pooled analysis (Figure 1).

Table 1 shows the risk of bias of the included RCTs assessed using the Cochrane Collaboration tool. Of the 14 studies included in the present systematic review, 13 (92.85%) presented a low risk of bias and 1 study (7.15%) presented an uncertain risk of bias because it did not present how the participants were randomized [20].

Table 2 presents the analysis of the methodological quality of the RCTs by the Jadad scale. The studies showed a high risk of bias (score ≤ 3). In the studies, randomization occurred in a simple way, despite having a satisfactory score in the description of sample loss and randomization. Double-blinding could improve the methodological quality of the studies.

Table 3 presents the years, countries, mean values and standard deviation of age, sample size, and BMD of participants of the studies included in the present systematic review. Interventions from the included studies consisted of a total of 1186 participants, with 691 participants in the experimental group (EG) and 495 in the control group (CG). It was found that the mean age of participants in the EG and CG of all studies was 68.4 years. The studies included in this review were developed in different countries, located on different continents. All participants were over 50 years of age. Publication years ranged from 1996 to 2021.

Table 4 shows the intervention and training volume of the studies. It was found that 12 studies had EG and CG, while 2 studies used only EG. The CG participants did not perform physical exercises, except for the studies of Dizdar et al. [21], García-Gomáriz et al. [24], and Paolucci et al. [28]. The EG participants performed strength, aerobic, balance, flexibility, and/or functional fitness exercises. The duration of the interventions ranged from 4 to 96 weeks, 20 to 60 min per training session, and a frequency of 2 to 5 sessions per week.

Table 5 presents the data on the evaluation and results of the included studies. The evaluation was divided between two and four moments according to each study. Functional fitness, BMD, and balance appeared more frequently in the included studies. Variables such as muscle strength, agility, quality of life, flexibility, pain assessment, and cardiorespiratory fitness were also analyzed and showed significant post-intervention increases (*p* < 0.05). The effect size (*d*) in the last column should be interpreted as follows: weak (<0.2), moderate (0.2 to 0.79), or strong (>0.8) [33].

Figure 2 shows the results of the meta-analyses of the studies that used the QUALEFFO-41 to evaluate the quality of life. The effect size was calculated by the standardized mean difference (SMD) with a confidence interval (CI) of 95%. When calculating the effect size, the negative sign means greater effects to the EG compared to the CG. In the forest plot, lines on the left side of the graph denote participants who received the multicomponent training intervention and presented significant positive changes compared to control participants. The average effect size of all RCTs is represented by the diamond and should be interpreted equally. There was a no significant difference in QUALEFFO-41 (95% CI: −2.06 to −0.69) with inconsistency I^2^ = 95% and *p*-value = 0.33.

Figure 3 presents the results of the meta-analyses of studies that used TUG for balance assessment. There was no significant difference in balance (95% CI: −1.41 to −0.50) with inconsistency I^2^ = 90% and *p*-value = 0.35.

Table 6 shows the level of evidence of the included studies, which was considered high, according to the GRADE tool. This means that there is moderate confidence in the estimated effect.

## 4. Discussion

The present study aimed to analyze the effects of multicomponent training on health-related variables of older women with osteoporosis. Increases in muscle strength, balance, cardiorespiratory fitness, and functional fitness were reported in the studies included in the present systematic review.

The included studies (*n* = 14) combined a minimum of two and a maximum of four different exercise types, involving strength, aerobic, balance, flexibility, and/or functional fitness training. The analysis of the 14 studies showed that older women with osteoporosis that practiced multicomponent training, with an average of 27.2 weeks, 2.6 sessions per week, and 45 min per session, improved strength, flexibility, QoL, BMD, balance, functional fitness, and reduced the risk of falls.

Different variables were analyzed in this systematic review. Balance was the most investigated variable, covering half (*n* = 7) of the included studies [7,20,21,23,25,26,31]. Muscle strength was evidenced in five studies [7,20,23,26,32]. Moreover, QoL was evaluated in six studies [7,21,22,28,30,31] and functional fitness was verified in six studies [7,27,29,30,32,34]. The frequency of falls was evaluated in three studies [23,25,27] and BMD was analyzed in four studies [24,26,29,32]. Flexibility was the least analyzed physical capacity among the included studies, being verified in two studies [22,23].

Burke et al. [20], Lord et al. [26], and Murtezani et al. [32] verified increases (*p* < 0.05) in isometric muscle strength of lower limbs in the knee, hip, and ankle flexion and extension movements with the application of the tests: knee extension, leg press, and back extensor strength. Murtezani et al. [32] reported increases (*p* < 0.05) in handgrip strength and lower limb strength. Filipović et al. [23] found an increase (*p* < 0.05) in lower limb muscle strength with the sit-to-stand test used to assess physical quality. Cardoso et al. [12] also reported increased muscle strength in upper and lower limb resistance exercises in a 12-week multicomponent program. However, Carter et al. [9] found no changes in lower limb muscle strength in the knee extension strength test.

Balance was the most analyzed variable in the studies included in this systematic review. Carter et al. [7] reported increases (*p* < 0.05) in balance using the Berg balance, while Murtezani et al. [32] found no changes in this variable using the same test. Burke et al. [20] and Halvarson et al. [25] reported improvements (*p* < 0.05) in balance through the COP velocity, directional control, balance performance, walking speed with a dual-task, fast walking speed, advanced lower extremity physical function, timed up and go (TUG), and Bretz stabilometer measurements. Lord et al. [26] and Carter et al. [7] found no differences (*p* > 0.05) in balance with the sway test and the composite balance score. Similarly, Dizdar et al. [21] and Filipović et al. [23] used the TUG test to assess balance and found no significant differences. Evstigneeva et al. [22] investigated flexibility and found no significant differences (*p* > 0.05). Nevertheless, increases in flexibility (*p* < 0.05) were reported by Olsen and Bergland [27] in the functional reach test.

Increases in QoL (*p* < 0.05) were observed in five studies, one [28] of them analyzed this variable with the Shortened Osteoporosis Quality of Life Questionnaire and four [21,22,30,31] used the 41-item Quality of Life Questionnaire of the European Foundation for Osteoporosis (QUALEFFO-41): pain, activities of daily mobility, jobs around the house, mobility, leisure social activities, general health perception, and mental function. On the other hand, Carter et al. [7] found no differences in the assessment of QoL in EG with the same instrument.

In the variable functional fitness, significant increases (*p* < 0.05) were reported in five studies [22,27,30,31,32], while the study by Carter et al. [7] did not present significant changes in this variable (*p* > 0.05). Different tests were used to assess functional fitness (eight-figure, test sit-to-stand weight transfer, six-minute walking test, maximum walking test, and functional reach test). However, the TUG test appeared more frequently in the evaluation of the functional fitness variable. Multicomponent training has been shown to be effective (*p* < 0.05) to improve the functional autonomy of older women [10], as well as resistance training with a frequency of two times and three times a week [6].

Few studies have investigated falls. The reduction in the frequency of falls (*p* < 0.05) was reported by Olsen and Bergland [27], Halvarsson et al. [25], and Filipović et al. [23] using the falls efficacy scale tests.

Lord et al. [26] used resistance training for 5 weeks and found no differences in BMD (*p* > 0.05) between the CG, but three studies [24,29,32] reported increases (*p* < 0.05) in BMD in the lumbar spine, forearm, and total BMD. A possible explanation may be the longer intervention time used in these studies, both with more than 40 weeks. The study of Borba-Pinheiro et al. [6] evaluated BMD, functional autonomy, muscle strength, and QoL in 52 postmenopausal women using different types of resistance training (RT), one performed twice a week (RT2) and the other performed three times a week (RT3). Both training programs (RT2 and RT3) showed positive results in 13 months of intervention when compared to the CG, using the Osteoporosis Assessment Questionnaire (OPAQ). Olsen and Bergland [27], with postmenopausal women using different types of exercises (water aerobics and judo) with 12 months of intervention, demonstrated that RT presented the best results (*p* < 0.05) for lumbar BMD, balance, and QoL (OPAQ) compared to other exercises and GC.

Of the 14 studies included in this systematic review, 4 studies were part of the meta-analysis. Evstigneeva et al. [22] and Stanghelle et al. [30] analyzed the quality of life using the QUALEFFO-41. These studies [22,30] showed favorable results (*p* < 0.05) with the multicomponent training intervention when compared to the CG (Figure 2). Additionally, two studies [22,23] evaluated the balance with the TUG test. Both of them showed improvements (*p* < 0.05) with the multicomponent training intervention when compared to the CG (Figure 3).

A limitation of the present systematic review to be highlighted was that some studies did not use the double-blind randomization methodological process. Furthermore, some studies investigated patients with and without fractures, which may interfere with the time and optimization of results. Other limitations to be considered are the different intervention protocols presented and the lack of data from some studies [22,25,27] regarding the confirmation of osteoporosis. The lack of patterns for the outcomes among the elected studies is another limitation. Moreover, there were a small number of studies included in the meta-analysis. Thus, these findings should be analyzed with caution when prescribing physical exercises for women with osteoporosis.

## 5. Conclusions

Physical exercise involving multicomponent training in women with osteoporosis can improve BMD, strength, flexibility, balance, functional fitness, and QoL, and reduce the risk of falls. Other types of physical exercise (aerobic, resistance, and flexibility) were presented in this review for this population. The results showed the importance of applying different forms of physical exercise as a treatment for osteoporosis in older women. Therefore, a physical exercise program that aims to stimulate different physical qualities in training sessions can promote musculoskeletal health and QoL in this population. Future studies are recommended to investigate body weight excess, due to low mobility, and rheumatic diseases, as they may be related to bone remodeling and the association of physical exercise in the health of older women with osteoporosis. Moreover, it is suggested to design and apply an intervention program of multicomponent exercise training for women with osteoporosis to determine if there are some positive effects on BMD.

## Figures and Tables

**Figure 1 ijerph-19-14195-f001:**
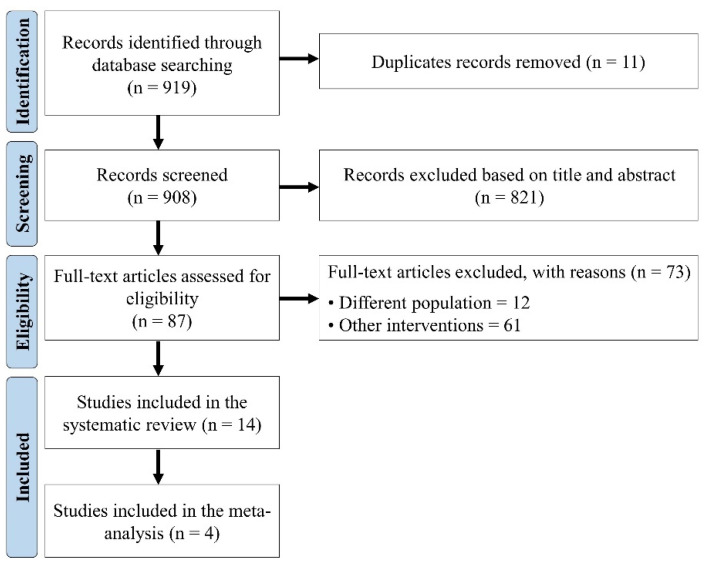
PRISMA flow diagram of study selection.

**Figure 2 ijerph-19-14195-f002:**
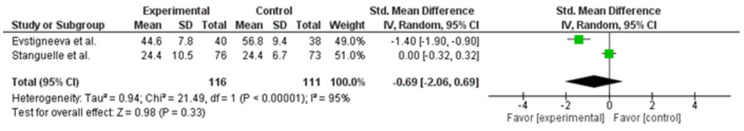
Forest plot (QUALEFFO-41) [22,30].

**Figure 3 ijerph-19-14195-f003:**
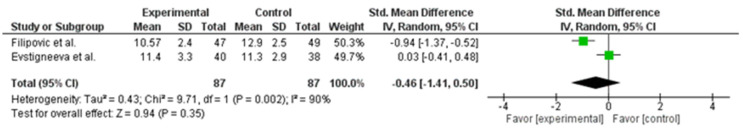
Forest plot (timed up and go) [22,23].

**Table 1 ijerph-19-14195-t001:** Risk of bias analysis for randomized controlled trials (Cochrane Collaboration tool).

Studies	1	2	3	4	5	6	7	Total
Burke et al. [20]	Uncertain	Low	Low	Low	Low	Low	Low	Uncertain
Carter et al. [7]	Low	Low	Low	Low	Low	Low	Low	Low
Dizdar et al. [21]	Low	Low	Low	Low	Low	Low	Low	Low
Evstigneeva et al. [22]	Low	Low	Low	Low	Low	Low	Low	Low
FilipoviĆ et al. [23]	Low	Low	Low	Low	Low	Low	Low	Low
Garcıa-Gomariz et al. [24]	Low	Low	Low	Low	Low	Low	Low	Low
Halvarsson et al. [25]	Low	Low	Low	Low	Low	Low	Low	Low
Lord et al. [26]	Low	Low	Low	Low	Low	Low	Low	Low
Murtezani et al. [23]	Low	Low	Low	Low	Low	Low	Low	Low
Olsen and Bergland [27]	Low	Low	Low	Low	Low	Low	Low	Low
Paolucci et al. [28]	Low	Low	Low	Low	Low	Low	Low	Low
Preisinger et al. [29]	Low	Low	Low	Low	Low	Low	Low	Low
Stanghelle et al. [30]	Low	Low	Low	Low	Low	Low	Low	Low
Nawrat-Szołtysik et al. [31]	Low	Low	Low	Low	Low	Low	Low	Low

1: randomization; 2: allocation of randomization; 3: blinding of participants; 4: blinding of the evaluators; 5: incomplete outcomes; 6: report on selective outcome; 7: other sources of bias.

**Table 2 ijerph-19-14195-t002:** Methodological quality assessment (Jadad scale).

Studies	1a	1b	2a	2b	3	Total Score
Burke et al. [20]	1	−1	0	0	1	1
Carter et al. [7]	1	1	0	0	1	3
Dizdar et al. [21]	1	1	0	0	1	3
Evstigneeva et al. [22]	1	1	0	0	1	3
FilipoviĆ et al. [23]	1	1	0	0	1	3
Garcıa-Gomariz et al. [24]	1	1	0	0	1	3
Halvarsson et al. [25]	1	1	0	0	1	3
Lord et al. [26]	1	1	0	0	1	3
Murtezani et al. [32]	1	1	0	0	1	3
Olsen and Bergland [27]	1	1	0	0	1	3
Paolucci et al. [28]	1	1	0	0	1	3
Preisinger et al. [29]	1	1	0	0	1	3
Stanghelle et al. [30]	1	1	0	0	1	3
Nawrat-Szołtysik et al. [31]	1	1	0	0	1	3

1a: randomized study; 1b: adequate randomization; 2a: double-blind study; 2b: proper blinding; 3: description of the sample loss.

**Table 3 ijerph-19-14195-t003:** Characteristics of the participants of the included studies.

Author	Year	Country	Age (Years)	EG (*n*)	CG (*n*)	Total (*n*)	BMD	
							(T-Score)	(g/cm^3^)
Lord et al. [26]	1996	Australia	EG: 71.7 ± 5.4 CG: 71.5 ± 5.5	90	89	179	NI	Lumbar spine: EG: 1.014 ± 0.2 CG: 0.965 ± 0.2 Femoral Neck: EG: 0.770 ± 0.1 CG: 0.742 ± 0.1 Trochanter EG: 0.689 ± 0.1 CG: 0.652 ± 0.1
Preisinger et al. [29]	1996	Austria	EG1: 62.6 ± 5.9 EG2: 60.9 ± 7.8 CG: 59.0 ± 8.0	EG1: 27 EG2: 34	31	92	NI	Distal forearm: EG1: 0.266 EG2: 0.269 CG: 0.305 Mid-forearm: EG1: 0.382 EG2: 0.383 CG: 0.424
Carter et al. [7]	2002	Colombia	EG: 69 ± 3.5 CG: 69.6 ± 3.0	40	40	80	NI	Total hip or lumbar spine ≤−2.5 SD
Burke et al. [20]	2010	Brazil	EG: 72.8 ± 3.6 CG: 74.4 ± 3.7	17	16	33	Lumbar spine: EG: −3.69 ± 0.83 CG: −3.53 ± 0.96	NI
Murtezani et al. [32]	2014	Switzerland	EG1: 60.68 ± 7.62 EG2: 59.78 ± 5.99	31	30	61	Lumbar spine: EG1: −3.04 ± 0.4 EG2: −3.10 ± 0.5	NI
Olsen and Bergland [27]	2014	Norway	EG: 70.4 ± 5.9 CG: 72 ± 5.6	47	42	89	NI	NI
Paolucci et al. [28]	2014	Switzerland	EG: 65.6 ± 5.8 CG: 65.6 ± 5.3	40	20	60	NI	Lumbar spine ≤−2.5 SD
Halvarsson et al. [25]	2014	Sweden	EG1: 76 ± 10 EG2: 77 ± 9 CG: 76 ± 10	EG1: 25 EG2: 18	26	69	NI	NI
Evstigneeva et al. [22]	2016	Russia	EG: 70.7 ± 8.1 CG: 67.6 ± 7.0	40	38	78	NI	NI
García−Gomáriz et al. [24]	2017	Spain	EG: 60.3 ± 5.4 CG: 56.5 ± 6.7	17	17	34	Femoral neck: −0.76 Lumbar spine: 1.93	NI
Dizdar et al. [21]	2017	Turkey	EG1: 57.87 ± 4.5 EG2: 59.86 ± 5.5 EG3: 60.91 ± 6.5	EG1: 25 EG2: 25 EG3: 25	–	75	Lumbar total: EG1: −2.44 ± 0.8 EG2: −2.62 ± 0.8 EG3: −2.54 ± 0.6 Femur neck: EG1: −1.67 ± 0.8 EG2: −1.85 ± 0.8 EG3: −1.97 ± 0.4 Femur total: EG1: −0.63 ± 1.2 EG2: −0.95 ± 0.9 EG3: −0.91 ± 1.1	NI
Nawrat−Szołtysik et al. [31]	2019	Poland	EG: 81.5 ± 10 CG: 81.5 ± 10	EG1: 23 EG2: 21 EG3: 23	24	91	≤1	NI
Stanghelle et al. [30]	2020	Norway	EG: 74.2 ± 5.8 CG: 74.7 ± 6.1	76	73	149	NI	Lumbar spine and femoral neck ≤ −2.5 SD
FilipoviĆ et al. [23]	2021	Serbia	EG: 64.40 ± 5.45 CG: 64.20 ± 5.08	47	49	96	Neck: −2.62 ± 0.72	Lumbar spine and femoral neck ≤ −2.5 SD

EG: experimental group; CG: control group; BMD: bone mineral density; NI: not informed; SD: standard deviation.

**Table 4 ijerph-19-14195-t004:** Study intervention data.

Study	Intervention	Duration (Weeks)	VT	
			DT (min)	FT (×/week)
Lord et al. [26]	EG: RT for upper limbs and balance CG: no physical exercises	5	60	4
Preisinger et al. [29]	EG1: regular RT, balance, motor, and postural coordination EG2: irregular RT, balance, motor, and postural coordination CG: no physical exercises	48	20	2
Carter et al. [7]	EG: RT, balance, postural exercises, and coordination CG: no physical exercises	20	40	2
Burke et al. [20]	EG: RT for lower limbs and balance CG: no physical exercises	8	30	2
Murtezani et al. [32]	EG1: RT, balance, and aerobic exercise (land) EG2: aerobic exercise and RT (water)	40	35	3
Olsen and Bergland [27]	EG: aerobic circuit training, balance, and flexibility CG: no physical exercises	12	60	3
Paolucci et al. [28]	EG: low-impact aerobics training, balance, and flexibility CG: aerobic training	24	60	3
Halvarsson et al. [25]	EG1: balance EG2: balance and aerobic training CG: no physical exercises	12	30–45	2
Evstigneeva et al. [22]	EG: RT for lower limbs and balance CG: no physical exercises	48	40	2
Dizdar et al. [21]	EG1: RT, balance, and coordination EG2: RT EG3: aerobic training	12–24	60	3
García-Gomáriz et al. [24]	EG: RT and high-impact training + calcium + vitamin D CG: high-intensity walk	96	60	2
Nawrat-Szołtysik et al. [31]	EG1: modified Sinaki exercises EG2: Nordic walking EG3: modified Sinaki exercises + Nordic walking CG: did not perform physical exercises	12	40	2
Stanghelle et al. [30]	EG: RT and balance CG: no physical exercises	24	60	2
FilipoviĆ et al. [23]	EG: RT, aerobic training, and balance CG: no physical exercises	4–24	50–60	5

EG: experimental group; CG: control group; VT: volume of training; DT: duration of training (each session); FT: frequency of training; RT: resistance training; min: minutes; ×/week: times per week.

**Table 5 ijerph-19-14195-t005:** Data from the variables analyzed and the results of the included studies.

Study	Evaluation	Results in the EG (*p* < 0.05)
Lord et al. [26]	Balance	EG: ↔ Sway (*d* = −0.30)
	BMD	EG: ↔ lumbar spine (*d* = 0.06); ↔ femoral neck (*d* = 0.08); ↔ trochanter (*d* = 0.04)
	Muscle strength	EG: ↑ quadriceps strength (*d* = 0.88)
Preisinger et al. [29]	BMD	EG1: ↑ distal forearm; ↔ mid-forearm EG2: ↓ distal forearm; ↓ mid-forearm
Carter et al. [7]	Balance	EG: ↔ composite balance score (*d* = 0.13)
	Functional fitness	EG: ↔ eight−figure (*d* = 0.33)
	Muscle strength	EG: ↔ knee extension strength (*d* = 0.09)
	QoL	EG: QUALEFFO−41: ↔ total score; ↔ social; ↔ general health perception; ↔ physical function; ↔ pain; ↔ mental state
Burke et al. [20]	Isometric muscle strength	EG: ↑ ankle flexion; ↑ knee extension; ↑ knee flexion
	Balance	EG: ↑ COP velocity; ↑ endpoint excursion; ↓ maximum excursion; ↓ directional control; ↑ stable surface/open eyes; ↓ stable surface/closed eyes; ↓ unstable surface/open eyes; ↓ unstable surface/closed eyes
Murtezani et al. [32]	Balance	EG1: ↔ BBS (*d* = 0.21) EG2: ↔ BBS (*d* = 0.02)
	Functional fitness	EG1: ↑ six-minute walking test (*d* = 0.96) EG2: ↑ six-minute walking test (*d* = 0.71)
	Muscle strength	EG1: ↑ quadriceps strength (*d* = 0.34); ↑grip strength (*d* = 0.76) EG2: ↑ quadriceps strength (*d* = 0.11); ↑grip strength (*d* = 0.02)
	BMD	EG1: ↑ BMD (*d* = 0.51) EG2: ↔ BMD (*d* = 0.07)
Olsen and Bergland [27]	Fall	EG: ↓ falls efficacy scale (*d* = −0.70)
	Functional fitness	EG: ↓ maximum walking speed (*d* = −0.40)
	Flexibility	EG: ↔ functional reach (*d* = 0.10)
Paolucci et al. [28]	Pain	EG1: ↓ VAS of pain (*d* = −1.58); ↓ McGill Pain Questionnaire (*d* = −0.60) EG2: ↑ VAS of pain (*d* = −2.31); ↑ McGill Pain Questionnaire (*d* = −2.31)
	QoL	EG1: ↑ Shortened Osteoporosis Quality of Life Questionnaire (*d* = 0.63) EG2: ↑ Shortened Osteoporosis Quality of Life Questionnaire (*d* = 0.88)
	Disability	EG1: ↓ Oswestry Disability Questionnaire (*d* = −0.63) EG2: ↓ Oswestry Disability Questionnaire (*d* = −0.88)
Halvarsson et al. [25]	Balance	EG: ↑ preferred speed single−task (*d* = 0.60); ↑ preferred speed dual-task (*d* = 1.00); ↔ error in the performance of the dual-task in percentage (*d* = 0.20); ↑ fast speed (*d* = 0.50); ↔ LLFDI: functional total (*d* = 0.40); ↔ upper extremity (*d* = 0.00); ↔ basic lower extremity (*d* = 0.20); ↑ advanced lower extremity (*d* = 0.60)
	Fall	EG: ↓ FES−I: ↑ one leg stance; ↑ modified figure−of−eight test time; ↑ physical activity; ↔ fear of falling; ↑ no percent; ↓ a little percent; ↓ quite a bit; ↓ very much; ↓ gait speed
Evstigneeva et al. [22]	QoL	EG: QUALEFFO−41: ↓ pain (*d* = −1.20); ↔ADL (0.10); ↓ mobility (*d* = −1.27), ↓ social function (*d* = −0.65); ↓ general health perception (*d* = −1.10); ↓ mental function (*d* = −0.48)
	Functional fitness	EG: ↔ Test weight−bearing/squat (*d* = 0.17); ↑ sit-to-stand weight transfer (*d* = −0.24); ↔ sit-to-stand left/right weight symmetry (*d* = −0.12); ↑ Tandem Walk and Sway test (*d* = −0.48); ↔ TUG (*d* = 0.03)
	Flexibility	EG: ↔ occiput−wall distance (*d* = 0.24)
Dizdar et al. [21]	Balance	EG: ↓ TUG (12th) (*d* = −0.33); ↑ BBS (12th) (*d* = 0.33)
	Pain	EG: ↓ VAS (12th) (*d* = −1.21)
	QoL	EG: QUALEFFO−41: ↓ total score (*d* = −0.43); ↓ pain (24th) (*d* = −0.43); ↔ physical function (*d* = −0.15); ↓ social function (*d* = −0.56); ↓ general health (*d* = −0.46); ↔ mental function (*d* = 0.06)
García−Gomáriz et al. [24]	BMD	EG: ↑ femoral neck (*d* = 0.37); ↔ lumbar spine (*d* = 0.41)
Nawrat−Szołtysik et al. [31]	Functional fitness	EG1 vs. EG2: ↑ number of steps and distance per day (*d* = 3.18); ↔ TUG; ↔ FRT EG2 vs. EG3: no differences (*p* > 0.05)
	QoL	EG1/EG2/EG3: QUALEFFO−41: ↔ pain, ↔ ADL; ↔ mobility; ↔ jobs around the house; ↔ mobility; ↔ leisure social activities; ↓ general health perception; ↓ mental function
Stanghelle et al. [30]	Functional fitness	EG: ↔ FRT (*d* = 0.39); ↓ four square step test (*d* = −0.32); ↔ grip strength right (*d* = −0.11); ↑ arm curl (*d* = −0.69); 30−s sit to stand (*d* = 0.44); ↔ TUG (*d* = −0.05); ↔ 6−min walking distance (*d* = 0.25)
	QoL	EG: HRQoL (QUALEFFO−41): ↓ FES−I (*d* = −0.13)
FilipoviĆ et al. [23]	Balance	EG: ↓ TUG (*d* = −0.63); ↑OLST (*d* = 0.76)
	Muscle strength	EG: ↓ STS (*d* = −0.80)
	Osteoporosis	EG: ↑ OKAT−S (*d* = 2.92)
	Fall	EG: ↓ FES−I (*d* = −1.15)

ADL: activities of daily living; BBS: Berg balance scale; BMD: body mass density; COP: center of pressure; *d*: effect size; FES-I: falls efficacy scale international; FRT: functional reach test; HRQoL: health-related quality of life; OKAT-S: osteoporosis knowledge assessment tool—short version; OLST: one leg stance test; OPAQ: osteoporosis assessment questionnaire; OQoLQ: Osteoporosis Quality of Life Questionnaire; QoL: quality of life; QUALEFFO-41: 41-item Quality of Life Questionnaire of the European Foundation for Osteoporosis; SF-36: 36-item short form health survey; STS: sit-to-stand test; TUG: timed up and go; VAS: visual analogue scale; LLFDI: late life disability and function instrument; ↑ increase; ↔ maintenance; ↓ reduction.

**Table 6 ijerph-19-14195-t006:** Level of evidence (GRADE).

Certainty Assessment							No. of Participants		Effect		Certainty	Importance
No. of Studies	Study Design	Risk of Bias	Inconsistency	Indirectness	Imprecision	Other Considerations	EG	CG	Relative (95% CI)	Absolute (95% CI)		
QoL (analyzed with QUALEFFO-41)												
2	RCTs	not serious	not serious	not serious	not serious	none	116	111	__	mean −1.09 highest (−2.06 lower to 0.69 higher)	⨁⨁⨁⨁ HIGH	Important
Balance (analyzed with TUG)												
2	RCTs	not serious	not serious	not serious	not serious	none	87	87	__	mean −0.46 highest (−1.41 lower to −0.50 higher)	⨁⨁⨁⨁ HIGH	Important

RCTs: randomized controlled trials; EG: experimental group; CG: control group; QoL: quality of life; QUALEFFO-41: 41-item Quality of Life Questionnaire of the European Foundation for Osteoporosis; TUG: timed up and go; CI: confidence interval; ⊕⊕⊕⊕: represents high confidence in the estimated effect.

## Data Availability

Not applicable.

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
