# Peer review of "Effects of Multicomponent Exercise Training on the Health of Older Women with Osteoporosis: A Systematic Review and Meta-Analysis"

_ijerph, 2022, doi:10.3390/ijerph192114195_

Round 1

Reviewer 1 Report (Previous Reviewer 1)

The authors have done a good job addressing most of my previous questions and comments. Significant changes to the introduction and discussion makes this is an improved version of the manuscript. There remains one last area where I would suggest improvement.

Multicomponent training was defined as a training modality that involves different physical capacities in the same exercise session and the retrieved studies combined a minimum of 2 different exercise types. Therefore, my suggestion is to add to the eligibility criteria the requirement of the multicomponent training defined as the training modality that combines a minimum of 2 different exercise types (strength, aerobic, balance, flexibility, and/or functional fitness).

Author Response

Dear reviewer,

The recommendations were made and are attached

Reviewer 2 Report (Previous Reviewer 2)

Dear authors, I appeciate yor effort, but for the future remains the recommendation to design and apply for a period of time a program of intervention in order to determine if there are some positive effects.

Author Response

Dear reviewer,

The recommendations were made and are attached

Reviewer 3 Report (New Reviewer)

Dear Author (s)

1. There are grammatical errors.

2. Why do not you add a meta-analysis?

3. Why did you search in Cochrane Library?

4. Were there any limitations during the search?

Author Response

Dear reviewer,

The recommendations were made and are attached

Round 2

Reviewer 3 Report (New Reviewer)

Dear Author (s)

I think without a meta-analysis, the study is not interesting for readers.

Author Response

Dear reviewer,

We appreciate the time you took to evaluate our manuscript and the valuable comments provided to improve our text. We provided the requested adjustments, which are highlighted in yellow in the manuscript file.

The satisfactory performance of the meta-analysis was only possible with the few studies that presented similar interventions and outcomes. (Borenstein M, Hedges LV, Higgins JPT, Rothstein HR. Introduction to Meta-Analysis, John Wiley & Sons, 2009; Souza, M. C. de. Métodos de Síntese e Evidência: Revisão Sistemática e Metanálise. INCA, 2015. Disponível em: < http://bvsms.saude.gov.br/bvs/publicacoes/inca/mirian_metodo_de_sinte se_e_evidencia.pdf>.)

This manuscript is a resubmission of an earlier submission. The following is a list of the peer review reports and author responses from that submission.

Round 1

Reviewer 1 Report

Effects of Multicomponent Exercise Training on the Health of Older Women with Osteoporosis: a Systematic Review

The study reviewed the health- related outcomes of multicomponent exercise training on elderly woman with osteoporosis. The technical aspect and rigor of the procedures are accurate and well-conducted for a systematic review. The main problem is the lack of an adequate eligibility criteria for selecting research involving osteoporosis. A clear and objective criteria for diagnosis of the dependent variable (osteoporosis) must be present in the eligibility criteria. However, there are no information about the method used for osteoporosis diagnosis. Additionally, is a cause of concern that only 4 from the 14 elected studies provided some kind of osteoporosis analysis.

Introduction

Probably, there are other systematic reviews on this interesting topic of public health interest. Therefore, what are the novelty and justification for the present systematic review?

I am missing a hypothesis here. As mentioned in the introduction, the exercise training is already well known to promote the maintenance of bone mass in elderly people. Why would be expected a different result in the elected studies?

The characterization of multicomponent exercise training is blurry to me. Additionally, the link between osteoporosis and multicomponent exercise training is not clear. What is the motivation to choose this modality of exercise training instead of others?

Methods

The study sounds technically well-conducted and the authors followed the PRISMA and PICO guidelines. Moreover, the study adopted good practices like the registration in the PROSPERO database, risk of bias and quality analysis.

The authors mentioned that there was not used filters for language. There were studies in other languages than English or Spanish which were retrieved by the search strategy employed?

The understanding of the strategy search would benefit from a table to facilitate the visualization of the mesh terms and boolean operators.

The eligibility criteria is the main problem of the study in my opinion. There was not adopted an objective criteria for osteoporosis diagnosis. The definition of multicomponent exercise training is vague and without filters for intensity, volume and frequency. The lack of pattern for the outcomes among the elected studies is another limitation.

Discussion and Conclusion

It is a hard task to compare and discuss the retrieved results due to the broad range of both exercise training protocols and observed outcomes. Therefore, the vague characterization of the multicomponent exercise training weakens its alleged improvement of health-related variables and recommendation for osteoporosis treatment.

Reviewer 2 Report

Dear authors

I reviewed your article and do not suggest it to be accepted for publishing.

In Introduction are just few data concerning the effects of the multicomponent training for osteoporosis.

Your analysis is made only on 14 studies, and 9 of them did not showed any values for BMD.

My recommendation is to design and apply for a period of time a program of multicomponent exercise training for women with osteoporosis in order to determine if there are some positive effects.
